# Assessing Changes in Land Use/Land Cover and Ecological Risk to Conserve Protected Areas in Urban–Rural Contexts

Isabelle D. Wolf [1,2], Parvaneh Sobhani [3] and Hassan Esmaeilzadeh [3,*]

1   Australian Centre for Culture, Environment, Society and Space, School of Geography and Sustainable Communities, University of Wollongong, Wollongong, NSW 2522, Australia
2   Centre for Ecosystem Science, University of New South Wales, Sydney, NSW 2052, Australia
3   Environmental Sciences Research Institute, Shahid Beheshti University, Tehran 1983969411, Iran
*   Correspondence: h_esmaeilzadeh@sbu.ac.ir

**Abstract:** Land use/land cover (LULC) changes in response to natural factors and human activities constitute a pressing issue for the conservation of Protected Areas in urban–rural landscapes. The present study investigated LULC changes in the Jajrud Protected Area (JPA) and the Kavdeh Wildlife Refuge (KWR) in the Tehran province, Iran, between 1989 and 2019. To inform ecological conservation measures for the JPA and KWR, LULC changes were identified and monitored using Landsat imagery from between 1989 and 2019. In addition, the landscape ecological risk (ER) was evaluated by conducting a landscape pattern index analysis. Then, the importance of different indicators affected by ER in these two PAs was assessed using the Delphi method, and expert opinions were solicited through a questionnaire. As for LULC changes in the JPA, high-density pasture declined the most over 1989–2019, from 38.6% (29,241 ha) to 37.7% (28,540 ha). In contrast, built-up areas increased the most, from 10.4% (7895 ha) in 1989 to 11.9% (9048 ha) in 2019. Water bodies also increased, from 0.88% (676 ha) in 1989 to 0.94% (715 ha) in 2019. In the KWR, cropland and gardens increased the most, from 2.14% (1647 ha) in 1989 to 3.4% (2606 ha) in 2019. Built-up areas also increased, from 0.05% (45 ha) in 1989 to 0.09% (75 ha) in 2019. Water bodies increased from 0.69% (538 ha) in 1989 to 0.71% (552 ha) in 2019. Finally, high-density pasture decreased the most, from 29.4% (22,603 ha) in 1989 to 28.5% (21,955 ha) in 2019. At the same time, the high and very high ER classes increased, more so in the JPA compared to the KWR. Finally, considering both LULC and ER changes, the Delphi method demonstrated that the greatest impacts occurred in the JPA. Various illegal economic and physical activities have created LULC changes and caused extensive destruction of ecosystems, posing a high ER in the study areas. The intensity of ER differs between the two PAs because of the varying distance from the metropolis, varying degrees of human activities, LULC changes, along with differences in legal restrictions of use. Aligned with the management plans of these areas, our research shows that it is necessary to develop land only within the designated zones to minimize the amount of ER. Various models of LULC changes have been presented, and a comparison of these models relating to the methodology and model effectiveness can help increase their accuracy and power of interpretation.

**Keywords:** land use/land cover (LULC) changes; ecological risk (ER) assessment; integrated system management; Delphi method; protected areas; Iran



## 1. Introduction

Global population growth and increased urbanization have created many risks for natural ecosystems, impacting landscape structures and functions [1–4]. In addition, human activities cause environmental pollution, increase ecological risk (ER) at various spatial scales, and drive land use/land cover LULC change on a small and large scale by changing people's lifestyles and expanding residential, industrial, and commercial centers [5–9]. LULC change reflects both the influence of human activities and of natural factors and, as a consequence, impacts natural ecosystems and increases ER [10,11]. In

particular, the use of Landsat images has attracted great interest for developing land use/land cover data due to free public access [12–15]. Geospatial techniques such as remote sensing are an efficient tool for developing land use classification maps. They vastly improve the selection of areas designated as agricultural, industrial, and/or urban land. Satellite remote sensing and GIS are the most common methods for the quantification, mapping, and detection of patterns of LULC because of their accurate georeferencing procedures, the digital format of data suitable for computer processing, and the repetitive data acquisition [16–19]. These techniques are more effective than conventional approaches because they offer high-resolution, informative, precise, and up-to-date information to investigate changes in landforms in a timely and cost-effective manner [20–23].

Protected Areas (PAs) in particular need to be managed in a way that LULC and ER are minimized, as they fulfill important functions for society to conserve natural ecosystems, ecosystem functions, habitats, and species. However, PAs also play a role in meeting the social and economic needs of communities, which can affect natural ecosystems [24]. Considering the importance of PAs, maintaining an ecological balance while delivering multiple ecosystem services is crucial [25,26]. Drastic LULC changes in PAs and the related increase in ER constitutes one of the most pressing environmental threats [27–29]. ER reflects the risk posed by human activities and natural environmental changes such as climate change, floods, and storms [30], which are both accounted for in ER assessments to determine the likelihood and extent of risk [31]. Assessing ER is challenged by the variability of landscapes and the intricate mosaics of different land cover types and uses they form. This variability is captured via the landscape pattern index. Both LULC change and the landscape pattern index influence ER [8,32] and are therefore most widely integrated in ER assessment methods [26,33,34].

The landscape pattern index indicates how landscape elements are spatially distributed and composed. In addition to showing the heterogeneity of the landscape, this index reflects the disturbance of the environment from LULC changes at various spatial and temporal scales. Multiple studies have used the landscape pattern index for their ER assessments. For instance, Xu et al. [35] undertook an ER assessment of the Pingshuo opencast coal mine and surrounding landscape in Western China. The results indicate that the high-risk area is expanding from southwest to northeast due to the operation of opencast mining, which has damaged land and landscapes severely. In other words, the risk posed from urban construction and mining lands in this area is relatively high, while forest areas and grasslands pose a relatively low risk. Consequently, ER changes there are mainly affected by the scale of mining operations, expansion of cities, conversion of agricultural lands into forests and meadows, land reclamation, and village relocation. In another study, Ke et al. [36] assessed ER using surface sediments from the Liaohe River PA in northeast China, which revealed heavy metal contamination: cadmium contributed most to the high to very high ER index values. Tuholske et al. [32] studied LULC changes in the Caribbean Islands for thirty years and assessed how urbanization related to mangrove loss. The results revealed that as urban land cover increased, mangrove plants decreased. Moreover, Li and Huang [37] studied landscape ER response to LULC change and uncovered a negative trend. Souza et al. [38] studied modeling of LULC change based on artificial neural networks for Santa Catarina/Brazil. Their results indicate that LULC changes have many different impacts, e.g., on climate change, biodiversity and ecosystem services, and soil quality, which, in turn, has implications for various landscape processes and functions. Tariq et al. [39] concluded for LULC changes in Pakistan that economic development, climate change, and population growth were the main driving forces. These various studies found that the landscape pattern index, which is a robust method for assessing LULC change spatially and temporally, can be used for ER assessments in PAs.

Although the evidence for the increasing destruction of natural ecosystems and increasing unsustainability of use of PAs is irrefutable, impacts of LULC changes on ER have not been holistically examined in PAs. Our study addresses this gap by presenting the case of two PAs located in the Tehran province of Iran, namely, the Jajrud Protected Area (Jajrud)

and the Kavdeh Wildlife Refuge (Kavdeh). In Tehran, LULC changes are substantive due to high population growth and rapid urbanization. Likely, both PAs are affected by LULC changes and increased ER because of livestock overgrazing, illegal construction, and unrestricted tourism development. Thus, a landscape ER assessment was conducted as an effective and important measure to inform conservation measures to protect biodiversity as well as valuable and rare biological species. LULC change was quantified using Landsat imagery and a remote sensing method evaluating imagery and data from 1989 to 2019. In addition, the Delphi method was used as a participatory technique to identify and assess ER. This is consistent with previous studies integrating the landscape pattern index into the analysis to assess ER and to study how it impacts PAs. Consequently, the main questions of this research are: (1) How much has LULC changed in the Jajrud PA and the Kavdeh WR from 1989 to 2019? (2) What is the status of ER in the studied areas? Finally, (3) what are the main impacts of ER in the study areas? These questions are underpinned by the following research hypotheses: (a) There is a relationship between LULC changes and ER in PAs. (b) Specifically, LULC changes increase ER in PAs.

## 2. Methods

### 2.1. Study Areas

The Jajrud PA and the Kavdeh WR were selected (Figure 1) for study because of the LULC changes caused by livestock overgrazing, illegal construction, unrestricted tourism development, and other factors that have led to increased ER [40]. Jajrud, with an area of about 75,670 ha, is located on the southern slope of the Alborz mountains in the Tehran province, which has a mountainous climate and altitudes ranging from 1000 to 2600 m. This area encompasses the two national parks Khojir and Sorkheh Hesar, covering an area of 8695 ha and 10,692 ha, respectively, that are known for their high levels of biodiversity. The area has a semiarid climate, with an average annual rainfall of 300 mm and temperatures of 11 degrees C. In terms of biodiversity, 517 plant species of 29 different genera have been identified (e.g., Artemisia sp., Bromus sp., Amygdalus sp., Zygophyllum sp., etc.). Wildlife species in this area include 38 mammal species, 118 bird species, 27 reptile species, 2 amphibian species, and 7 fish species. The Asiatic Mouflon (*Ovis orientalis*) is a known indicator animal species [41,42]. Kavdeh offers one of the major wildlife habitats among the PAs in this area. It is located about 160 km at east of Tehran and covers an area of 76,900 ha. In terms of biodiversity, 405 plant species and 159 animal species have been identified [42].

### 2.2. General Framework and Data

Figure 2 shows a flow diagram of the methodology we applied: LULC changes were identified and monitored in the Jajrud and Kavdeh using Landsat imagery from between 1989 and 2019. In addition, the landscape ER was evaluated by conducting a landscape pattern index analysis. Finally, we adopted the Delphi method to assess the importance of different indicators affected by ER in PAs, soliciting expert opinions through a questionnaire. Table 1 presents a list of dimensions, variables, and indicators affected by ER in PAs, as collated through a literature review. This list consists of 3 dimensions, namely, physical–environmental, socio-cultural, and economic–institutional, as well as 26 variables and 35 indicators. According to Table 1, ER can affect multiple variables, such as habitats, biodiversity, ecosystems, environments, wildlife species, and vegetation.

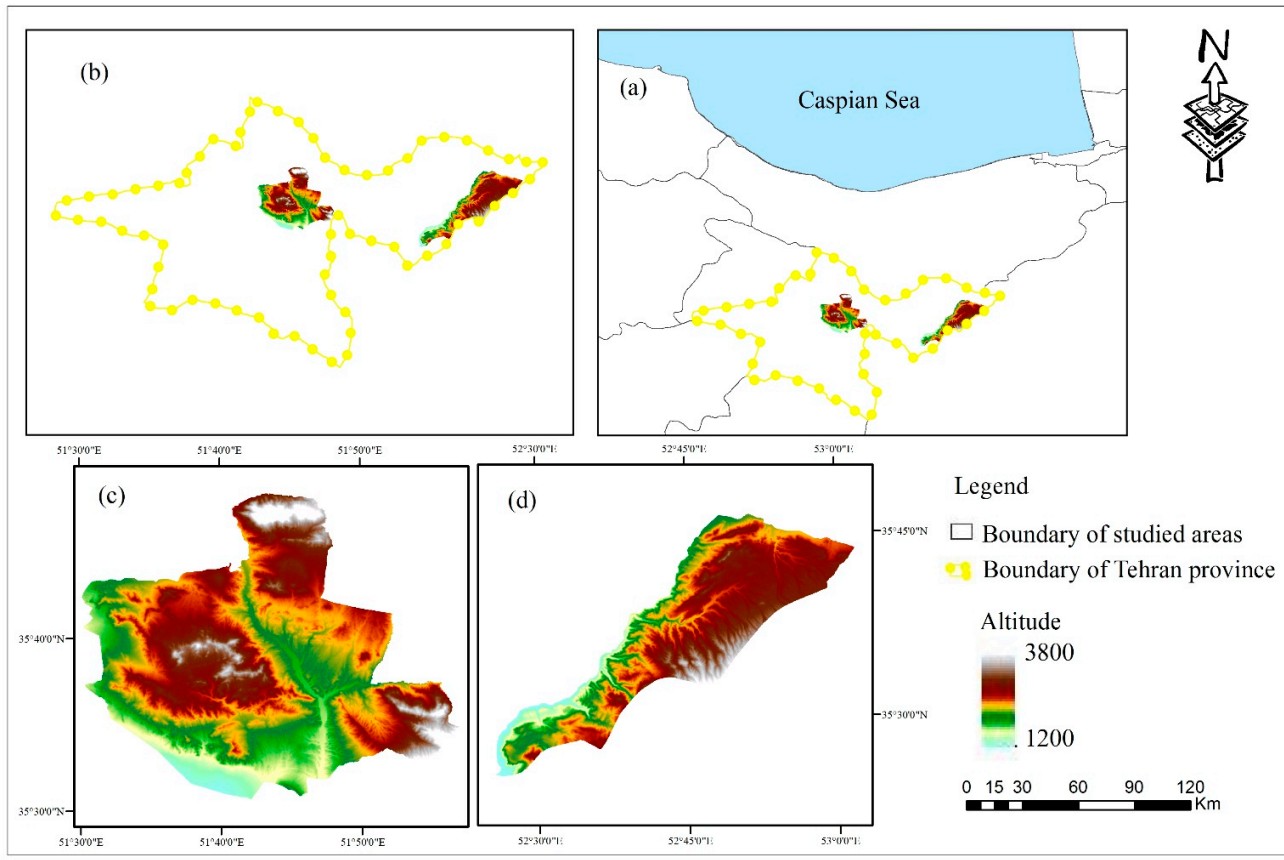

**Figure 1.** Location of the study areas: (**a**) Iran, (**b**) Tehran province, (**c**) Jajrud PA, and (**d**) Kavdeh WR.

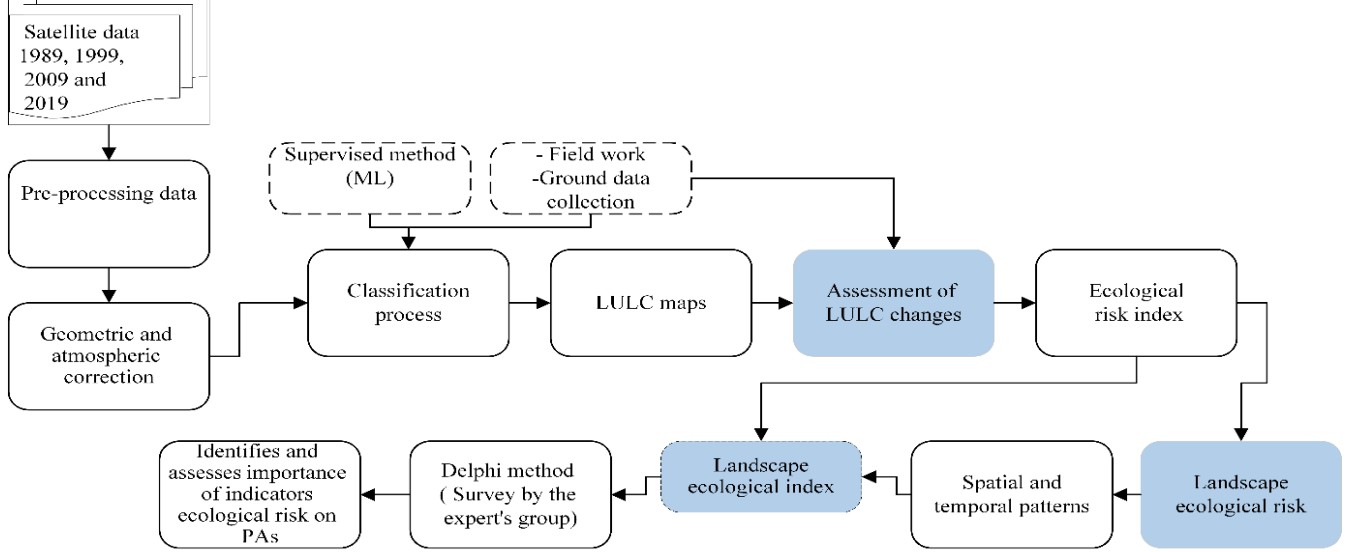

**Figure 2.** The process of the research.

**Table 1.** List of dimensions, variables, and indicators affected by ecological risk (ER) in Protected Areas.

| Dimensions | Variables | Indicators | References |
|---|---|---|---|
| Physical–environmental | Habitat | Habitat integrity | [11,26,35,43–54] |
| | | Unique habitat | |
| | | Source-sink dynamics | |
| | Biodiversity | Biodiversity | |
| | Ecosystem | Natural ecosystem | |
| | | Ecosystem functions (flow of matters, energy and information, etc.) | |
| | Environment | Environmental pollutions | |
| | Wildlife species | Extinction of valuable biological species | |
| | Vegetation | Vegetation density (such as high-density pasture, low-density pasture, forest, agricultural land, and orchards) | |
| | | Overgrazing of livestock | |
| | Climate | Climate change | |
| | Natural resources | Reservoirs of groundwater aquifers and surface water | |
| | | Soil erosion (sedimentation and soil fertility) | |
| | Landscape | Landscape structure (such as patches, corridors and matrix) | |
| | | Landscape fragmentation | |
| | | Landscape vulnerability | |
| | | Ecological flows (genetic information) | |
| | | Edge effects (due to isolation habitats) | |
| | Environmental hazards | Abrupt environmental crises (such as storms, floods, earthquakes, etc.) | |
| Socio-cultural | Tourism attractions | Tourism attractions (natural, historical and cultural, man-made attractions) | |
| | Educational services | Educational programs | |
| | Density | Population density (tourists, visitors, local communities) | |
| | Satisfaction | Social satisfaction | |
| | Security | Food security | |
| | Infrastructures | Illegal infrastructure | |
| | Health | Human health | |
| | Institutional elements | Control and monitoring | |
| | Rules and regulations | Legal restrictions | |
| Economic–institutional | Constructions | Illegal build up | |
| | Employment | Employment opportunities | |
| | Income | Income of communities | |
| | Prices | Prices of estate and commodities | |
| | Plans and projects | Illegal plans and projects | |
| | Agricultural | Cultivated lands | |
| | Tourism | Tourism activities | |

*2.3. Data Collection and Classification of Satellite Data*

To assess how LULC has changed over time, LULC types were developed from L5-TM (21 April 1989), L5-TM (21 April 1999), L7-ETM+ (21 April 2009), and L8 and OLI-TIRS (21 April 2019) of the US Geological Survey (USGS, https://earthexplorer.usgs.gov/, accessed on 10 May 2022). Since the classification of satellite images is the most important part when extracting changes, supervised methods are mainly used for classification to achieve the greatest accuracy [55]. Following geometric and atmospheric corrections, LULC of PAs were classified based on the maximum likelihood algorithm. Accordingly, LULC in Jajrud was classified as built-up, water body, cropland and orchard, high-density pasture, low-density pasture, and planted forest. In Kavdeh, LULC was classified in five classes, including built-up, water body, cropland and orchard, high-density pasture, and

low-density pasture. Using training samples for each class, 600 samples were examined from the sampled collection, 400 samples were examined for algorithm training, and 200 samples were evaluated for classification evaluation. Finally, using a set of ground truth samples, the classification accuracy was compared with the training samples for each LULC class (300 pixels). During this study, two images were used for each year, as the stage in planting and harvesting changes throughout the year, which affects the cropland and orchard classifications. Principal component analysis (PCA) (see [56] for details), was used to summarize layers and merge them into several bands. To classify roads as part of built-up areas, the vector layer was converted into a raster image with a 30-m pixel.

*2.4. Ecological Risk Index*

In order to calculate the ecological risk index (ERI), a sampling time interval system was employed. The study areas were divided into units for the ER assessment of 10 km × 10 km [57]. Then, based on the landscape loss index in each year, the ERI was calculated, and results were assigned to the central pixel of the assessment areas. The ERI was calculated using a landscape disturbance index and a landscape vulnerability index to describe severe disturbances and vulnerabilities of PAs. Therefore, this index elucidates the degree of disturbance and vulnerability of ecosystems in Jajrud and Kavdeh and the relationship between landscape patterns and ER based on LULC changes. These indexes were selected considering the ecological importance of landscapes, the conditions in the study areas, and other related studies [26,43,58–60]. The calculation formula is as follows:

$$ERI_K = \sum_{i=1}^{N} \frac{AK_i}{AK} \times D_i \times V_i \tag{1}$$

where $K$ = area unit, $i$ = landscape type, $N$ = total number of landscape components, $AK_i$ = the area of landscape $i$ in the $k$ sample area, $AK$ = the total area of sample $K$, $D_i$ = the landscape disturbance index of type $i$, and $V_i$ = the landscape fragility index. The higher the ERI value, the higher the ecological risk.

Table 2 shows the ecological meaning and calculation of the landscape index. The landscape disturbance index describes the degree of human disturbance that occurs in a development process caused by different landscape components and leads to changes in natural resources and the environment [61]. On the other hand, the landscape separation index shows the degree to which patches are dispersed [62]. Moreover, the landscape separation index was used to elucidate the impact of external interference on the structure of environmental networks. The fractal dimension index of the landscape emphasizes regularity of the geometry of patches and indicates the complexity of patch shape [58]. For this index, we measure the degree of irregularity and intensification of landscape fragments, which is used to express the degree of morphological change in the landscape due to external interferences such as human activities and LULC change [63]. Thus, here, the landscape disturbance index was used to describe the extent of human disturbance to the landscape. Hence, based on human activities and unplanned development in PAs, the landscape Fragmentation index ($F_i$), Separation index ($S_i$), and Fractal Dimension index ($FD_i$) were selected to construct the landscape disturbance index in a manner that is consistent with previous studies [2,59]. The fragmentation and separation indexes directly reflect the landscape shape, among which the Fragmentation index elucidates more ER and the Fractal Dimension index indirectly examines the landscape shape. Moreover, the landscape Vulnerability index depends on the ability of ecosystem and landscape components to withstand external interference. Therefore, various ecosystems have different abilities to withstand changes due to external interference and disturbance [34,54,57,64,65]. In this study, each of the ER indexes has been applied according to local conditions and characteristics of the studied areas. Finally, to determine the sensitivity coefficients for each LULC class, the landscape Vulnerability index ($V_i$) was used according to the degree

of vulnerability (ranging from high to low) to external interference resulting from the development of human activities (Table 2).

**Table 2.** Ecological meaning and calculation of the landscape index (adapted from Zhang et al., 2019).

| Index | Symbol | Computation | Ecological Meaning of Index |
|---|---|---|---|
| Landscape fragmentation | $F_i$ | $F_i = \frac{n_i}{A_i}$ | $F_i$ is employed to elucidate the fragmentation degree of the landscape which transitions from continuous whole patches to complex discontinuous patches caused by natural or human disturbances. As the value increases, the landscape ecosystem's stability will decrease. In the equation, $n_i$ is the number of patches of landscape type $i$, and $A_i$ is the area of landscape type $i$. |
| Landscape separation | $S_i$ | $S_i = \frac{1}{2}\sqrt{\frac{n_i}{A_i}} \times \frac{A}{A_i}$ | In a landscape type, $S_i$ indicates how well patches are separated from one another. As values increase, the spatial distribution of the landscape type $i$ becomes more complex, and the separation degree is higher. This equation describes the number of patches of the landscape type $i$ by $n_i$ and the total area of type $i$ by $A$. |
| Landscape fractal dimension | $FD_i$ | $FD_i = \frac{2\ln\left(\frac{P_i}{4}\right)}{\ln Ai}$ | The value range of $FD_i$ is 1–2. The larger the value, the more complex the shape of the landscape patches. When $FD_i < 1.5$, the patch shape is relatively simple; when $FD_i = 1.5$, the patch is in a Brownian random motion state, with poor stability; when $FD_i > 1.5$, the patch shape is complex. In the equation, $P_i$ is the perimeter of the landscape type $i$. |
| Landscape disturbance | $D_i$ | $D_i = aF_i + bS_i + cFD_i$ | $D_i$ identifies the level of interference between different landscapes based on the level of human exploitation. $a + b + c$ equals one, where $a + b + c$ represents the weight. |
| Landscape vulnerability | $V_i$ | Jajrud PA<br><br>6- Water body<br>5- Cropland and garden<br>4- High-density pasture<br>3- Low-densitypasture<br>2- Planted forests<br>1- Built-up<br><br>Kavdeh wildlife refuge<br><br>5- Water body<br>4- Cropland and garden<br>3- High-density pasture<br>2- Low-density pasture<br>1- Built-up | Depending on the type of landscape, $V_i$ reflects how sensitive it is to disturbance from external factors. The degree of succession is determined by the stage of the landscape ecosystem. In the present study, landscape types are categorized according to land use/land cover and vulnerability (from high to low), based on previous research and the characteristics of the study areas. |

### 2.5. Delphi Method

The Delphi method was used to investigate the structure of the impacts. This method is used to achieve a comprehensive understanding of change processes following projects and events over time and helps with analyzing and informing decision making. The Delphi method adheres to a structured process for collecting and classifying knowledge solicited from experts and different stakeholders [29]. The assumption is that experts have in-depth knowledge about the subject in study. The Delphi method sets the conditions for researchers to achieve a theoretical consensus even from heterogeneous expert viewpoints and where not all data and information is accessible [66–69]. As shown in Table 1, the impacts of ER in PAs were investigated in the physical–environmental, socio-cultural, and economic–institutional dimensions along with the identified variables and indicators (Table 1). We invited 35 experts to participate in the identification and assessment of impacts using a series of questionnaires. Experts were selected from academics and other

staff of the Department of Environment specialized in various fields such as environmental engineering, biodiversity, geography, zoology, biology, and landscape design. Experts rated impacts of ER on a 5-point scale (1 = very low, 2 = low, 3 = moderate, 4 = high, and 5 = very high). Mean ratings were calculated and statistically analyzed. To validate the results, experts met in 3 rounds. In the first round, indicators extracted from a literature review were presented to the experts, and they were asked to express their viewpoints about them. In the second round, the indicators presented in the first round and those suggested additionally were evaluated by the experts. Finally, in the third round, the total number of indicators obtained from the previous two rounds were presented to the experts, and they were asked to express their final viewpoints to reach a consensus. Therefore, in the validation stage, to ensure accuracy of viewpoints, the experts reviewed a final list of indicators. Table 3 shows participation rates of experts that decrease marginally over the three rounds, from 35 to 33 and then to 32 experts.

**Table 3.** Experts who participated in the Delphi study.

| Category | Science Field | Round 1 | Round 2 | Round 3 |
|---|---|---|---|---|
| Academics and employees of the Department of Environment | Environmental engineering | 5 | 5 | 4 |
| | Biodiversity | 9 | 8 | 7 |
| | Zoology | 7 | 7 | 8 |
| | Biology | 5 | 5 | 5 |
| | Landscape designing and planning | 6 | 5 | 5 |
| | Geography | 3 | 3 | 3 |
| Total | | 35 | 33 | 32 |

## 3. Results

### 3.1. LULC Changes

This study monitored LULC changes in two Pas, including the Jajrud and the Kavdeh, using Landsat imagery from 1989 to 2019. The envisaged accuracy of classification was obtained according to Table 4. The results revealed that the overall accuracy was high in terms of efficiency, and that an acceptable level was reached. As for LULC changes in Jajrud, high-density pasture area declined the most over 1989–2019, from 38.6% (29,241 ha) to 37.7% (28,540 ha) (Table 5). In contrast, built-up areas increased the most, from 10.4% (7895 ha) in 1989 to 11.9% (9048 ha) in 2019. In the same area, planted forest decreased from 2.3% (1754 ha) in 1989 to 2.2% (1676 ha) in 2019, and low-density pasture also decreased, from 45.43% (34,380 ha) in 1989 to 44.85% (33,938 ha) in 2019. In Jajrud, cropland and gardens increased, albeit slightly, from 2.27% (1724 ha) in 1989 to 2.31% (1753 ha) in 2019. Water bodies also increased, from 0.88% (676 ha) in 1989 to 0.94% (715 ha) in 2019 (Figure 3). In Kavdeh, cropland and gardens increased the most, from 2.14% (1647 ha) in 1989 to 3.38% (2606 ha) in 2019. Built-up areas also increased, from 0.05% (45 ha) in 1989 to 0.09% (75 ha) in 2019. Water bodies increased from 0.69% (538 ha) in 1989 to 0.71% (552 ha) in 2019. In the same area, high-density pasture decreased the most, from 29.39% (22,603 ha) in 1989 to 28.55% (21,955 ha) in 2019. Low-density pasture also decreased, from 67.7% (52,066 ha) in 1989 to 67.2% (51,711 ha) in 2019 (Figure 4).

**Table 4.** Overall LULC classification accuracy achieved from using Landsat imagery for two Protected Areas in the Tehran province, Iran, for the time period of 1989–2019.

| The Cases of Study | Year | Prepared LULC Map |
|---|---|---|
| | | Overall Accuracy |
| Jajrud | 1989 | 0.87 |
| | 1999 | 0.92 |
| | 2009 | 0.85 |
| | 2019 | 0.97 |
| Kavdeh | 1989 | 0.92 |
| | 1999 | 0.84 |
| | 2009 | 0.91 |
| | 2019 | 0.96 |

**Table 5.** Changes of LULC classes in two Protected Areas in the Tehran province, Iran, over 1989–2019.

| | Jajrud | | | | | | | | | |
|---|---|---|---|---|---|---|---|---|---|---|
| Year | 1989 | | 1999 | | 2009 | | 2019 | | Variation 1989–2019 | |
| | Area | | Area | | Area | | Area | | Area | |
| LULC | ha | % | ha | % | ha | % | ha | % | ha | % |
| Built-up | 7895 | 10.43 | 7997 | 10.56 | 8386 | 11.08 | 9048 | 11.95 | 1153 | 1.52 |
| Water body | 676 | 0.88 | 682 | 0.9 | 698 | 0.92 | 715 | 0.94 | 39 | 0.06 |
| Cropland and garden | 1724 | 2.27 | 1738 | 2.28 | 1744 | 2.30 | 1753 | 2.31 | 29 | 0.04 |
| High-density pasture | 29,241 | 38.64 | 29,212 | 38.60 | 29,150 | 38.52 | 28,540 | 37.71 | −701 | −0.93 |
| Low-density pasture | 34,380 | 45.43 | 34,298 | 45.32 | 33,998 | 44.92 | 33,938 | 44.85 | −442 | −0.58 |
| Planted forests | 1754 | 2.31 | 1743 | 2.30 | 1694 | 2.23 | 1676 | 2.21 | −78 | −0.1 |
| Sum total | 75,670 | 100 | 75,670 | 100 | 75,670 | 100 | 75,670 | 100 | — | — |
| | Kavdeh | | | | | | | | | |
| Built-up | 45 | 0.05 | 52 | 0.06 | 61 | 0.07 | 75 | 0.09 | 30 | 0.04 |
| Water body | 538 | 0.69 | 542 | 0.70 | 547 | 0.71 | 552 | 0.71 | 14 | 0.02 |
| Cropland and garden | 1647 | 2.14 | 1923 | 2.50 | 2154 | 2.80 | 2606 | 3.38 | 959 | 1.24 |
| High-density pasture | 22,603 | 29.39 | 22,512 | 29.27 | 22,335 | 29.04 | 21,955 | 28.55 | 648 | −0.84 |
| Low-density pasture | 52,066 | 67.70 | 51,870 | 67.45 | 51,802 | 67.36 | 51,711 | 67.24 | 355 | −0.46 |
| Sum total | 76,900 | 100 | 76,900 | 100 | 76,900 | 100 | 76,900 | 100 | — | — |

*3.2. Changes in the Landscape Indexes*

As for changes in landscape pattern indexes in the Jajrud from 1989 to 2019, the $F_i$, $S_i$, $D_i$, and $FD_i$ indexes increased for high-density pastures, low-density pastures, and planted forests, while they decreased for built-up, water bodies, and croplands and gardens. These results explain the increase in separation and distance of patches and also the decrease in connection of habitats. In the Kavdeh, the $F_i$, $S_i$, $D_i$, and $FD_i$ indexes increased for high-density pastures, which explains the increase in distance and separation of patches, habitat fragmentation, and the unsustainable use of ecosystems. Moreover, the $F_i$, $S_i$, $D_i$, and $FD_i$ indexes decreased for built-up areas, bare land, and low-density pastures, which indicates a decrease in the connection of patches. As the results demonstrate, habitat fragmentation and the distance of patches increased, while the size of patches decreased.

During 1989–2019, $F_i$, $S_i$, $D_i$, and $FD_i$ increased the most for high-density pastures, which elucidates an increase in the distance of pasture patches (or an increased number of pasture patches in a smaller size). The $V_i$ index elucidates the sensitivity of various types of landscapes to external disturbances, so that its degree is related to the succession stage of the landscape ecosystem (Table 6). Reflective of the characteristics of the study area, types of landscapes have been identified from high to low based on LULC classes and their vulnerability. Among these classes, water bodies are landscape components that are most exposed to external disturbance from human activities. Consequently, the highest sensitivity coefficient is assigned to this class. Conversely, the lowest sensitivity coefficient was found for the other LULC classes in terms of their vulnerability, ranging from high to low, including croplands and gardens, pasture lands (high-density and low-density pasture), planted forests, and finally, built-up areas. Hence, vulnerability coefficients were classified from high to low as follows: (1) Jajrud: 6 for water bodies, 5 for croplands and gardens, 4 for high-density pastures, 3 for low-density pastures, 2 for planted forests, and 1 for built-up areas; and (2) Kavdeh: 5 for water bodies, 4 for croplands and gardens, 3 for high-density pastures, 2 for low-density pastures, and 1 for built-up areas.

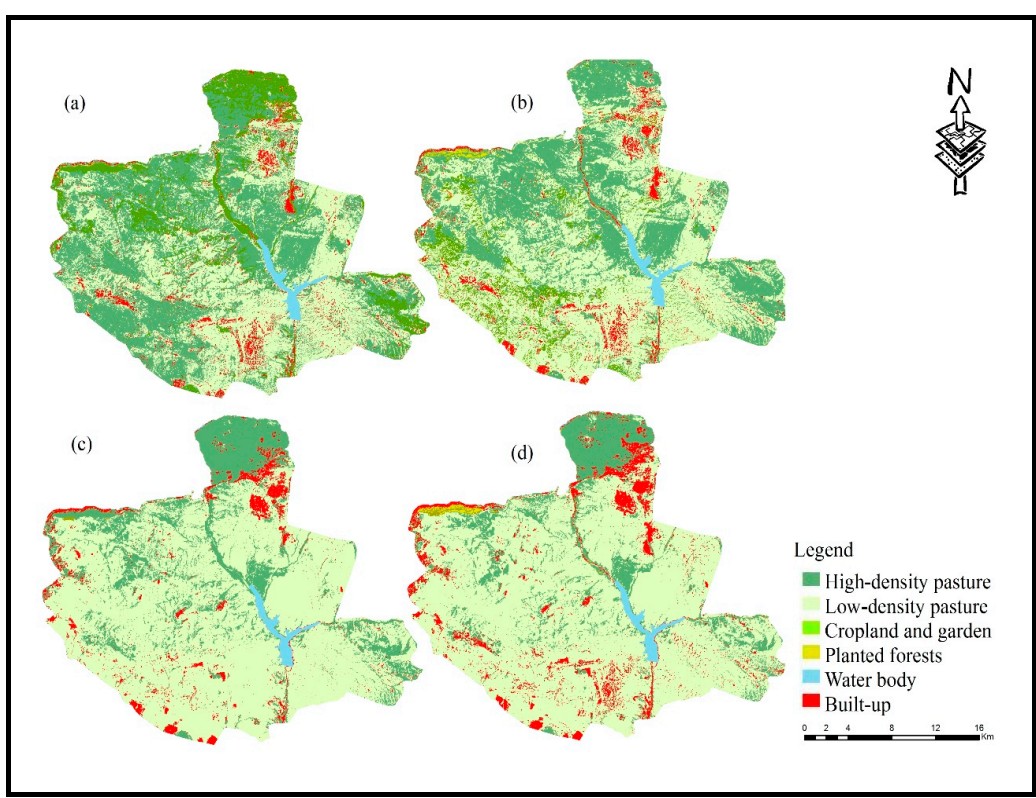

**Figure 3.** LULC changes for the Jajrud Protected Area, Iran: (**a**) 1989, (**b**) 1999, (**c**) 2009, (**d**) 2019.

*3.3. ER Index (ERI) Changes*

ER was classified into five classes, including very high, high, medium, low, and very low (Table 7, Figure 5). For the Jajrud in the period from 1989–1999, ER grade ranged from very high (0.29%), high (0.22%), medium (−0.14%), low (−0.23%) to very low risk (−0.07%). From 2009 to 2019, the ER grade ranged from very high (0.37%), to high (0.28%), medium (−0.24%), low (−0.12%), and very low (−0.15%), which reflects clear changes over time. In Kavdeh in the period from 1989–1999, the ER grade ranged from very high (0.22%) to high (0.19%), medium (0.12%), low (0.09%), and very low risk (−0.15%). Likewise, from 2009 to 2019, the ER grade ranged from very high (0.27%) to high (0.18%), medium (0.15%), low (0.05%), and very low (−0.24%) levels. Thus, across 1989 to 2019, the very high and high ER classes increased. Accordingly, from 1989 to 1999, high ER classes increased, while from 2009 to 2019, low-ER classes decreased.

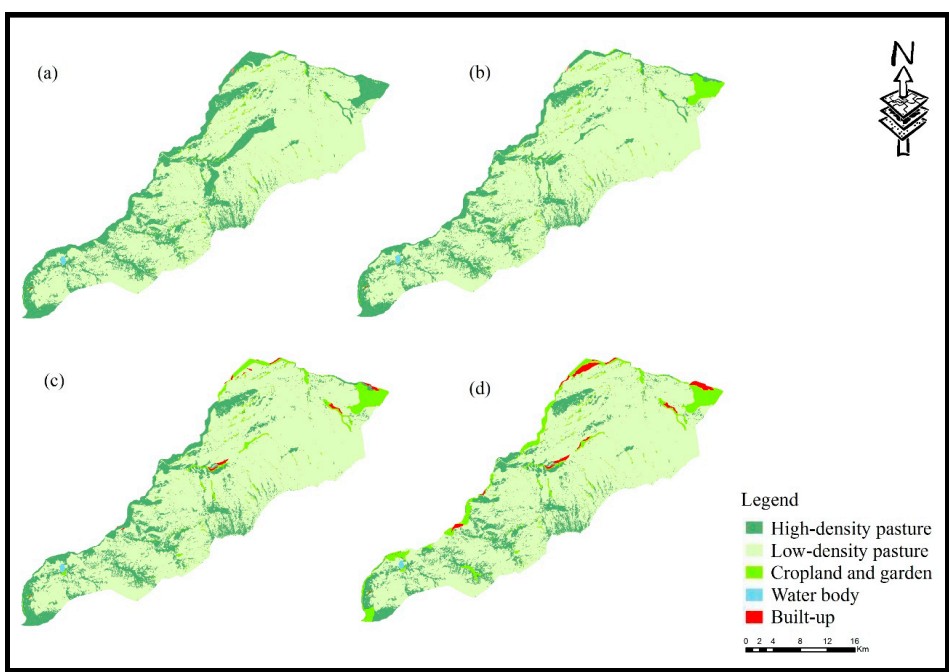

**Figure 4.** LULC changes for the Kavdeh Wildlife Reserve, Iran: (**a**) 1989, (**b**) 1999, (**c**) 2009, (**d**) 2019.

**Figure 5.** ER changes in the study areas: (**a**) Jajrud, (**b**) Kavdeh, (**1**) 1989–1999, (**2**) 2009–2019.

**Table 6.** Changes of landscape pattern indexes in two Protected Areas in the Tehran province, Iran, over 1989–2019.

| | | | | Jajrud | | | | | |
|---|---|---|---|---|---|---|---|---|---|
| **LULC** | **Year** | **Area (ha)** | **Area (%)** | **Patches** | $F_i$ | $S_i$ | $FD_i$ | $D_i$ | $V_i$ |
| Built-up | 1989 | 7895 | 10.43 | 1205 | 1.628 | 0.487 | 1.48 | 0.3488 | 0.0562 |
| | 1999 | 7997 | 10.56 | 1150 | 1.621 | 0.470 | 1.42 | 0.3482 | 0.0554 |
| | 2009 | 8386 | 11.08 | 1108 | 1.588 | 0.462 | 1.36 | 0.3461 | 0.0548 |
| | 2019 | 9048 | 11.95 | 1095 | 1.566 | 0.456 | 1.22 | 0.3455 | 0.0532 |
| Water body | 1989 | 676 | 0.88 | 95 | 0.351 | 0.097 | 1.36 | 0.2155 | 0.0468 |
| | 1999 | 682 | 0.9 | 87 | 0.338 | 0.085 | 1.32 | 0.2152 | 0.0457 |
| | 2009 | 698 | 0.92 | 85 | 0.322 | 0.078 | 1.23 | 0.2148 | 0.0438 |
| | 2019 | 715 | 0.94 | 72 | 0.315 | 0.062 | 1.21 | 0.2147 | 0.0426 |
| Cropland and garden | 1989 | 1724 | 2.27 | 456 | 0.590 | 0.1848 | 1.45 | 0.1857 | 0.0321 |
| | 1999 | 1738 | 2.28 | 434 | 0.582 | 0.1832 | 1.37 | 0.1713 | 0.0318 |
| | 2009 | 1744 | 2.30 | 423 | 0.578 | 0.1825 | 1.35 | 0.1686 | 0.0308 |
| | 2019 | 1753 | 2.31 | 412 | 0.562 | 0.1805 | 1.29 | 0.1542 | 0.0297 |
| High-density pasture | 1989 | 29,241 | 38.64 | 1561 | 0.068 | 1.2751 | 1.63 | 0.6452 | 0.0675 |
| | 1999 | 29,212 | 38.60 | 1587 | 0.075 | 1.2768 | 1.68 | 0.6502 | 0.0682 |
| | 2009 | 29,150 | 38.52 | 1595 | 0.084 | 1.2792 | 1.72 | 0.6521 | 0.0691 |
| | 2019 | 28,540 | 37.71 | 1621 | 0.092 | 1.2804 | 1.78 | 0.6538 | 0.0698 |
| Low-density pasture | 1989 | 34,380 | 45.43 | 1365 | 0.349 | 0.9765 | 1.52 | 0.4562 | 0.0171 |
| | 1999 | 34,298 | 45.32 | 1385 | 0.356 | 0.9782 | 1.64 | 0.4567 | 0.0176 |
| | 2009 | 33,998 | 44.92 | 1414 | 0.367 | 0.9820 | 1.67 | 0.4572 | 0.0182 |
| | 2019 | 33,938 | 44.85 | 1450 | 0.378 | 0.9851 | 1.73 | 0.4580 | 0.0185 |
| Planted forests | 1989 | 1754 | 2.31 | 1342 | 0.165 | 0.1796 | 1.56 | 0.3253 | 0.0254 |
| | 1999 | 1743 | 2.30 | 1351 | 0.170 | 0.1806 | 1.58 | 0.3268 | 0.0261 |
| | 2009 | 1694 | 2.23 | 1368 | 0.176 | 0.1822 | 1.66 | 0.3272 | 0.0268 |
| | 2019 | 1676 | 2.21 | 1375 | 0.188 | 0.1842 | 1.67 | 0.3288 | 0.0270 |
| | | | | Kavdeh | | | | | |
| **LULC** | **Year** | **Area (ha)** | **Area (%)** | **Patches** | $F_i$ | $S_i$ | $FD_i$ | $D_i$ | $V_i$ |
| Built-up | 1989 | 45 | 0.05 | 145 | 0.3455 | 0.9871 | 1.48 | 0.2956 | 0.0456 |
| | 1999 | 52 | 0.06 | 136 | 0.3423 | 0.9862 | 1.32 | 0.2942 | 0.0450 |
| | 2009 | 61 | 0.07 | 128 | 0.3415 | 0.9850 | 1.26 | 0.2935 | 0.0442 |
| | 2019 | 75 | 0.09 | 138 | 0.3402 | 0.9846 | 1.18 | 0.2918 | 0.0438 |
| Water body | 1989 | 538 | 0.69 | 82 | 0.326 | 0.085 | 1.28 | 0.2235 | 0.0432 |
| | 1999 | 542 | 0.70 | 75 | 0.318 | 0.072 | 1.22 | 0.2231 | 0.0427 |
| | 2009 | 547 | 0.71 | 63 | 0.308 | 0.066 | 1.18 | 0.2225 | 0.0422 |
| | 2019 | 552 | 0.71 | 55 | 0.295 | 0.057 | 1.14 | 0.2217 | 0.0416 |
| Cropland and garden | 1989 | 1647 | 2.14 | 78 | 0.346 | 0.165 | 1.55 | 0.1745 | 0.0351 |
| | 1999 | 1923 | 2.50 | 64 | 0.332 | 0.154 | 1.42 | 0.1742 | 0.0346 |
| | 2009 | 2154 | 2.80 | 56 | 0.327 | 0.145 | 1.37 | 0.1736 | 0.0332 |
| | 2019 | 2606 | 3.38 | 45 | 0.314 | 0.138 | 1.25 | 0.1728 | 0.0325 |
| High-density pasture | 1989 | 22,603 | 29.39 | 163 | 0.5975 | 3.2476 | 1.66 | 0.4475 | 0.0163 |
| | 1999 | 22,512 | 29.27 | 167 | 0.5982 | 3.2488 | 1.71 | 0.4482 | 0.0172 |
| | 2009 | 22,335 | 29.04 | 168 | 0.6721 | 3.2515 | 1.75 | 0.4498 | 0.0182 |
| | 2019 | 21,955 | 28.55 | 174 | 0.6708 | 3.2541 | 1.78 | 0.4512 | 0.0185 |
| Low-density pasture | 1989 | 52,066 | 67.70 | 125 | 0.3132 | 1.1526 | 1.23 | 0.1826 | 0.0232 |
| | 1999 | 51,870 | 67.45 | 128 | 0.3137 | 1.1538 | 1.28 | 0.1832 | 0.0238 |
| | 2009 | 51,802 | 67.36 | 133 | 0.3145 | 1.1542 | 1.34 | 0.1838 | 0.0244 |
| | 2019 | 51,711 | 67.24 | 142 | 0.3152 | 1.1548 | 1.43 | 0.1845 | 0.0254 |

**Table 7.** Percent distribution of ecological risk in two Protected Areas in the Tehran province, Iran.

| Studied Areas | Year | Ecological Risk Grade (%) | | | | |
|---|---|---|---|---|---|---|
| | | Very High | High | Medium | Low | Very Low |
| Jajrud | 1989 | 8.03 | 27.23 | 32.92 | 23.01 | 9.32 |
| | 1999 | 8.32 | 27.45 | 32.78 | 22.78 | 9.25 |
| | 2009 | 8.21 | 28.59 | 31.82 | 22.76 | 8.38 |
| | 2019 | 8.58 | 28.81 | 31.58 | 22.64 | 8.23 |
| | 1989–1999 | 0.29 | 0.22 | −0.14 | −0.23 | −0.07 |
| | 2009–2019 | 0.37 | 0.28 | −0.24 | −0.12 | −0.15 |
| Kavdeh | 1989 | 7.76 | 27.45 | 32.91 | 22.41 | 8.64 |
| | 1999 | 7.98 | 27.64 | 33.03 | 22.50 | 8.49 |
| | 2009 | 7.96 | 28.46 | 32.10 | 22.19 | 8.64 |
| | 2019 | 8.23 | 28.64 | 32.25 | 22.24 | 8.40 |
| | 1989–1999 | 0.22 | 0.19 | 0.12 | 0.09 | −0.15 |
| | 2009–2019 | 0.27 | 0.18 | 0.15 | 0.05 | −0.24 |

*3.4. Management Plans of PAs*

Since the study was undertaken in PAs, any natural, social, and economic development is prohibited there. PA management policies posit strict regulations. The Department of the Environment of the Tehran Province is legally responsible for the management of PAs in this area and therefore any physical or economic activities need to be licensed and monitored by this agency. The provisioned management plans underpin the protective status of the land instead of a developmental approach. Accordingly, nine zones have been identified in the Jajrud including a strict nature reserve, a protected zone, a recovery zone, a buffer zone, and a common protection zone, covering 65,446 ha dedicated mainly to protecting this land. Conversely, the extensive-use zone, intensive-use zone, special-use zone, and multiple-use zone, that cover 10,224 ha, allow for small-scale development [70]. Following these zoning standards, 10,224 ha of the Jajrud can be developed on a small scale (Table 8, Figure 6). As long as the LULC changes in this area occur on a small scale, the ER will be minimized. Considering that the Kavdeh area has recently been defined as a wildlife refuge, a management plan has not yet been prepared for it [71].

**Table 8.** Identified zones in the Jajrud Protected Area.

| Zones | Area | Developmental Approach | | Protective Approach | |
|---|---|---|---|---|---|
| | | ha | % | ha | % |
| Strict Nature Reserve | 11,311 | | - | 11,311 | 15 |
| Protected zone | 25,693 | | - | 25,693 | 34 |
| Extensive use zone | 753 | 753 | 1 | - | - |
| Intensive use zone | 20 | 20 | 1 | - | - |
| Recovery zone | 14,576 | | - | 14,576 | 19 |
| Special use zone | 2 | 2 | 1 | - | - |
| Buffer zone | 7785 | | - | 7785 | 10 |
| Multiple use zone | 9449 | 9449 | 11 | - | - |
| Common protection zone | 6081 | | - | 6081 | 8 |
| Sum total | 75,670 | 10,224 | 14 | 65,446 | 86 |

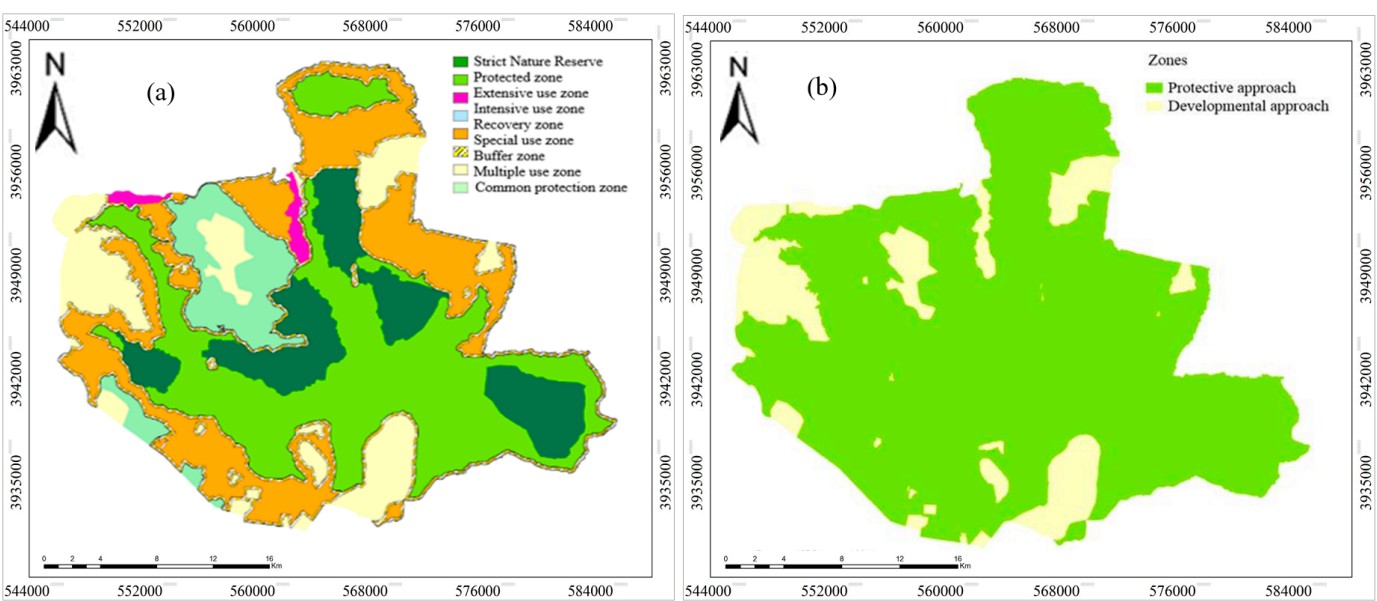

**Figure 6.** Zoning of the Jajrud Protected Area with (**a**) identified zones, (**b**) with an either protective or developmental approach to management.

### 3.5. ER Impacts

Table 9 summarizes the socio-demographics of the experts responding to the questionnaires delivered in the Delphi method. Respondents included 66% males and 34% females, with the majority being aged between 30 and 40 years old (51.35%). The respondents had a high level of education: 17.14% held a bachelor' degree, 34.28% held an M.Sc. degree, and 48.57% held a Ph.D. degree. Among them, 71.42% of respondents were academic professors, and 28.57% were employees of the Department of Environment of Iran. As for their scientific specialization, 20% of respondents were from the field of environmental engineering, 23% were in biodiversity, 20% in zoology, 20% in biology, 11% in landscape design, and 6% in geography (Table 9).

### 3.6. Assessment of Indicators Affected by ER

For the assessment of indicators affected by ER, various methods were mixed to collate an initial list of impact indicators (Table 10), including a review of theoretical and empirical literature and expert opinion. An additional 13 indicators were added to the list by the experts (Table 1). This included 11 indicators in the physical–environmental dimension, namely, ecosystem services, species' diet, species' migration, behavioral patterns, resilience and biological capacity of species, plant pests and diseases, landscape and environmental desirability, landscape heterogeneity, aesthetic quality of landscape, ecological connectivity (among habitats), spatial element patterns and structure (size, shape, number, type, composition, etc.). Two indicators were added to the economic–institutional dimension, namely, employment in different sectors (agriculture, industries, mines, services, etc.) and the number of tourists.

Table 5 presents a list of indicators affected by ER. In Jajrud, the physical–environmental indicator most affected includes "landscape fragmentation", with a score of 4.68. Conversely, the least-affected indicator is the "growth of plant pests and diseases", with a score of 1.28. In the socio-cultural dimension, the indicator thought to be most affected is the "growth of illegal infrastructures", with a score of 2.95, while the "increase of educational programs" only scored 2.18. Finally, the indicator in the economic–institutional dimension that was thought to be most affected is the "growth of illegal build-up", with a score of 3.24. The least-affected indicator in this dimension was "control and monitoring", with a score of 2.36. In Kavdeh, in the physical–environmental dimension, the indicator of "loss of habitat integrity" was thought to be most affected, with a score of 4.32. Likewise, the

lowest value was related to "expansion of plant pests and diseases", with a score of 1.12. In the socio-cultural dimension, the "reduction of social satisfaction" was thought to be most affected, with a score of 2.66; not far behind, but with the lowest score of 2.10, was the indicator relating to the "development of educational programs". Finally, the indicator most affected in the economic–institutional dimension was the "decrease in tourism activities", with a score of 3.03, and the least-affected indicator was the "development of illegal plans and projects", with a score of 2.28. Thus, the greatest mean ER was found for the physical–environmental dimension, while the lowest mean ER was found for the socio-cultural dimension.

**Table 9.** Socio-demographic profile of the respondents participating in the Delphi study component to assess indicators affected by ecological risk in two Protected Areas in the Tehran province, Iran.

| Characteristics | Frequency (N) N = 35 | Percentage |
|---|---|---|
| Gender | | |
| Male | 23 | 66 |
| Female | 12 | 34 |
| Age | | |
| Less than 30 years' old | 2 | 5.71 |
| 30–40 years' old | 18 | 51.35 |
| 40–50 years' old | 11 | 31.28 |
| 50+ years old | 4 | 11.40 |
| Education | | |
| Bachelor's degree | 6 | 17.14 |
| M.Sc. degree | 12 | 34.28 |
| Ph.D. degree | 17 | 48.57 |
| Work status | | |
| Academics | 25 | 71.42 |
| Employees of the Department of Environment | 10 | 28.57 |
| Scientific field | | |
| Environmental engineering | 7 | 20 |
| Biodiversity | 8 | 23 |
| Zoology | 7 | 20 |
| Biology | 7 | 20 |
| Landscape designing and planning | 4 | 11 |
| Geography | 2 | 6 |

**Table 10.** Average and values of expert ratings (1 = lowest, 5 = highest) to assess the extent to which indicators are affected by ecological risk in two Protected Areas in the Tehran province, Iran.

| Dimensions | Variables | Indicators | Jajrud | | Kavdeh | |
|---|---|---|---|---|---|---|
| | | | Average | Value | Average | Value |
| Physical–environmental | Habitat | Loss of habitat integrity | 3.35 | 4.66 | 3.11 | 4.32 |
| | | Reduction of unique habitats | | 3.30 | | 3.18 |
| | | Disturbance of source-sink dynamics | | 3.15 | | 2.89 |
| | Biodiversity | Loss of biodiversity | | 3.45 | | 3.36 |
| | Ecosystem | Destruction of natural ecosystem | | 2.56 | | 2.48 |
| | | Disturbance of ecosystem functions (flow of matters, energy, information, etc.) | | 3.88 | | 3.72 |
| | Environment | Reduction of ecosystem services | | 3.05 | | 2.92 |
| | | Increase in environmental pollutions level | | 2.88 | | 2.52 |
| | Wildlife species | Disturbance of species' diet | | 3.54 | | 2.94 |
| | | Increase in species' migration level | | 2.95 | | 2.87 |
| | | Disturbance of behavioral patterns | | 2.77 | | 2.56 |
| | | Decrease in resilience level and biological capacity of species | | 3.84 | | 3.22 |

**Table 10.** *Cont.*

| Dimensions | Variables | Indicators | Jajrud | | Kavdeh | |
|---|---|---|---|---|---|---|
| | | | Average | Value | Average | Value |
| Physical–environmental | Vegetation | Increase in extinction level of valuable biological species | | 3.66 | | 3.45 |
| | | Decrease in vegetation density level (such as high-density pasture, low-density pasture, forest, agricultural land, and gardens) | | 2.55 | | 2.26 |
| | | Growth of overgrazing of livestock | | 2.32 | | 2.24 |
| | | Increase in plant pests and diseases | | 1.28 | | 1.12 |
| | Climate | Increase in climate change degree | | 1.92 | | 1.72 |
| | Natural resources | Reduction of the groundwater aquifers and surface water reservoirs | | 3.34 | | 2.97 |
| | | Increase in soil erosion level (sedimentation and soil fertility) | | 3.15 | | 3.03 |
| | Landscape | Reduction of landscape and environmental desirability | | 3.92 | | 3.55 |
| | | Decrease in landscape heterogeneity level | | 3.38 | | 2.94 |
| | | Reduction of aesthetic quality of landscape | | 2.24 | | 1.98 |
| | | Disturbance of landscape structure (such as patches, corridors, and matrix) | | 4.55 | | 4.31 |
| | | Increase in landscape fragmentation level | | 4.68 | | 4.12 |
| | | Increase in landscape vulnerability level | | 4.18 | | 3.98 |
| | | Reduction of ecological connectivity (among habitats) | | 4.23 | | 4.30 |
| | | Disturbance of ecological flows (genetic information) | | 4.32 | | 4.18 |
| | | Growth of edge effects (due to isolation habitats) | | 3.68 | | 3.32 |
| | | Disturbance of patterns and spatial elements' structure (size, shape, number, type, composition, etc.) | | 4.46 | | 4.25 |
| | Environmental hazards | Growth of abrupt environmental crises (such as storm, flood earthquake, etc.) | | 2.66 | | 2.58 |
| Socio-cultural | Tourism attractions | Growth of tourism attractions (natural, historical and cultural, man-made attractions) | 2.59 | 2.68 | 2.46 | 2.56 |
| | Educational services | Increase in educational programs level | | 2.18 | | 2.06 |
| | Density | Decrease in population density level (tourists, visitors, local communities) | | 2.22 | | 2.10 |
| | Satisfaction | Decrease in social satisfaction level | | 2.44 | | 2.66 |
| | Security | Reduction of food security | | 2.88 | | 2.50 |
| | Infrastructures | Growth of illegal infrastructures | | 2.95 | | 2.71 |
| | Health | Decrease in human's health level | | 2.82 | | 2.68 |
| Economic–institutional | Institutional elements | Growth of control and monitoring systems | 2.81 | 2.36 | 2.7 | 2.45 |
| | Rules and regulations | Increase in legal restrictions level | | 2.63 | | 2.58 |
| | Constructions | Growth of illegal build up | | 3.24 | | 2.83 |
| | Employment | Reduction of employment opportunities | | 2.92 | | 2.77 |
| | | Reduction of employment level in different sectors (agriculture, industries, mines, services, etc.) | | 2.86 | | 2.63 |
| | Income | Decrease in communities' income volume | | 3.10 | | 2.92 |
| | Prices | Growth of estate and commodities prices | | 2.94 | | 2.90 |
| | Plans and projects | Growth of illegal plans and projects | | 2.83 | | 2.28 |
| | Agricultural | Decrease in cultivated lands volume | | 2.94 | | 2.86 |
| | Tourism | Reduction of tourism activities | | 2.74 | | 3.03 |
| | | Decrease of tourists' number | | 2.35 | | 2.45 |

## 4. Discussion

The current study detected LULC change in the Jajrud Protected Area and the Kavdeh Wildlife Reserve, Iran, using Landsat imagery from between 1989 to 2019. Moreover, a landscape pattern index analysis was conducted to assess ER. Finally, this study investigated the impacts of ER on various indictors along the physical–environmental, socio-cultural, and economic–institutional dimensions, using the Delphi method. The results showed that LULC and ER change in the Jajrud was mainly driven by an increase in built-up areas, which at the same time led to a decrease in high-density pastures. Lobbying has led to different parts of this PA being assigned to various organizations for financial exploitation which exacerbates the rapid destruction of ecosystems and habitats in the area. This issue has also undermined the authority in decision-making held by the Department of the Environment of Tehran province as a custodian of this land. Several issues, such as the conservation budget and logistical shortcomings, must be analyzed to determine whether the Department of the Environment of Tehran province is accomplishing its goal to preserve natural resources and biodiversity in PAs [72–74]. Another important issue increasing the LULC changes in this area is the increase of human activities, particularly dams, roads, residential complexes, factories, industrial and mining activities, canalization, and gas pipes. The inability of the Department of the Environment of Tehran to monitor and manage the area, along with the strong influence of various governmental stakeholders, has intensified human activities in the area. Furthermore, for some time the deterioration of PAs was and continues to not be detected because of a lack of monitoring [75]. In addition, the lack of a concerted cooperation among different organizations and the Department of the Environment of Tehran to protect this area has caused extensive destruction and unsustainable use. Finally, policies and management plans are not well implemented in the Jajrud which is impacted on by the proximity to the Tehran metropolis, with its rapid urbanization, illegal tourism activities, and the focus on economic development. These findings have been confirmed by other studies [4,37–39,76,77], which evidence LULC changes and destructive environmental impacts from human activities.

In contrast, in the Kavdeh, LULC changes are not significant yet, likely because of the greater distance to the Tehran metropolis, the lack of expansion of industrial and mining activities, and the limited development of other human activities and construction. The relatively minor LULC changes that were noted for this area relate to overgrazing, insufficient monitoring, and the uncontrolled development of ecotourism activities. These findings have been confirmed by other studies [78]. Likewise, in the Jajrud, ER has occurred during 2009–2019 more than during 1989–1999 due to development of uncontrolled physical and economic activities, such as the expansion of transportation infrastructures, the increasing demand for recreation and tourism in pristine and natural areas, population growth, and the expansion of cities. In the Kavdeh, from 1989 to 2019, the ER also increased due to a variety of causes. One of the main reasons is the overgrazing of livestock held by nomads, with the number of livestock exceeding the carrying capacity of pastures. This has decreased the proportion of high-density pasture compared to low-density pasture in this area. Other important issues include uncontrolled hunting and livestock overgrazing. In previous studies, overgrazing was reported as the most important reason for pasture degradation [79,80]. Further reasons include the uncontrolled movement and activities of visitors for hunting, walking, mountaineering, rock climbing, and off-road vehicle driving. Several famous tourist attractions are located in this area. The large number of tourists visiting this area causes ecosystem degradation, landscape fragmentation, and unsustainable environmental use. Monitoring is currently too insufficient to control the number of nomads and tourists flocking into the area. This situation has and continues to have many challenging consequences for this region, including changes in land use/land cover, the reduction of vegetation density and quality, mortality or displacement of birds and other wildlife, the disturbance of the ecological balance of the region, and there like. Our findings are consistent with those presented by others [43,81–84]. Similarly, previous

research indicates that economic development, population growth, tourism activities, and physical development of cities, have caused LULC changes and ER in PAs [79,80,82,83].

The results further indicate that the LULC and ER changes in the Jajrud and the Kavdeh is giving rise to the destruction of valuable biological habitats. due to the low level of monitoring, failure of coordinated conservation efforts of different organizations, the lack of an integrated management system, etc. The continuation of these changes without proper planning and establishment of managerial strategies will further ER and destruction of natural resources in future years. Finally, the results of the Delphi method reveal the negative impacts of the ER in PAs, which are frequently manifested along the physical–environmental dimension through the decrease in habitat integrity, the increase in landscape vulnerability, habitat fragmentation, disturbance of patterns and spatial element structure (size, shape, number, type, composition, etc.), and the disturbance of landscape structure (such as patches, corridors, and matrix). Another major issue is the lack of coordination among organizations and orchestration of conservation efforts by the Department of the Environment of the Tehran Province to manage and monitor these issues. Poor infrastructure and uncontrolled development have exacerbated the situation. The Jajrud was more affected compared to the Kavdeh due to unimpeded development and illegal activities by various groups, and the high percentage of LULC changes in this area.

In the present study, issues relating to LULC and ER changes in PAs were raised. Managers of PAs in Iran do not strictly reinforce regulations and laws especially where they cause conflict with socio-economic demands [85]. The methodology we have presented is capable of quantifying LULC changes, and impacts of human activities on natural ecosystems which is particularly important to monitor unacceptable changes in PAs. Quantifying ER changes can help managers protect these areas and achieve environmental sustainability. Hence, the results of this study emphasize the importance of estimating the effect of LULC changes on ER and their impacts on a broad range of indicators. In the future the focus could be directed towards specific ecological features of the study areas such as differences in structures and environmental processes. Finally, while our methodological approach helps with the quantitative and statistical calculation of LULC and ER changes in PAs and other natural areas, and the identification of factors leading to ecological destruction, and the formulation of conservation policies, one limitation relates to its origins in human health studies and their different purpose to assess change and risk. Thus, as more studies such as ours are being developed more insights should emerge that confirm the reliability of this approach.

## 5. Conclusions

Investigating LULC changes and ER in PAs can help prevent the destruction of ecosystems. LULC models are effective at monitoring changes and alerting conservation agencies and other stakeholders to unacceptable forms of use. This study examined ER based on LULC change within two Protected Areas in Iran: The Jajrud Protected Area and the Kavdeh Wildlife Reserve. Our research showed how LULC changes lead to increased ER along physical–environmental, socio-cultural, and economic–institutional dimensions. Underestimating the effect of LULC changes on ER poses a serious threat to Protected Areas. Even though the Jajrud and the Kavdeh are legally protected, various illegal economic and physical activities have created LULC changes and caused extensive destruction of ecosystems, leading to high ER. The intensity of the ER differs between the two Protected Areas due to factors such as the varying distance from the Tehran metropolis, varying degrees of human activities, and LULC change, along with differences in legal restrictions imposed by the Department of the Environment of Iran. Our research revealed that an increased economic and physical activities in the study areas has decreased the integrity of habitats. LULC and ER changes in the Jajrud were mainly driven by increased areas of build-up land, while high-density pastures decreased. In contrast, in the Kavdeh, LULC changes are not yet significant, likely because of the greater distance to the Tehran metropolis, the lack of expansion of industrial and mining activities, and the limited development of other human

activities and construction. Thee relatively minor LULC changes that were noted were mainly caused by overgrazing, insufficient monitoring, and the uncontrolled development of ecotourism activities. The lack of adequate monitoring and management, low levels of public awareness, further the lack of participatory conservation action by stakeholders, and haphazard forms of exploitation, has seen a drastic increase in LULC changes and environmental degradation over the last decade. To manage LULC changes and ER, makes it necessary to develop an integrated management system to coordinate organizational conservation efforts better.

The custodian of these areas is the Department of Environment of the Tehran Province. Hence, any physical or economic activities should be licensed and monitored only by this Department. Moreover, the reinforcement of strict regulations is necessary through management plans and zoning approaches. Monitoring of human activities that lead to LULC change and increase ER such as unplanned development, illegal construction, unplanned tourism activities, and overgrazing is essential too. Our study demonstrated how to assess changes in LULC and ER and their impacts on a broad range of indicators. This type of information is critical to inform managers and decision makers to review LULC changes and manage ER in Protected Areas. While the current study assessed ER using various landscape indices, other landscape indices and methods of risk assessment can be adopted to predict impacts from ER in Protected Areas. PAs require adequate planning and management; thus, by analyzing past changes and predicting the probability of future changes, the findings of this study will help managers in monitoring and controlling unacceptable change in PAs. Future research directions have been proposed, including the study of various reactions of organisms to chemicals using our ER assessment approach and formulating optimal scenarios for each organism; further a comparison of various LULC models to assess their effectiveness, simulation of future LULC and ER changes in PAs, calculating ecosystem services threatened by LULC changes in PAs, and a meta-analysis of strategies controlling LULC and ER changes in PAs.

**Author Contributions:** Conceptualization, I.D.W. and H.E.; methodology, H.E. and I.D.W.; software, P.S. and H.E.; validation, P.S. and H.E.; formal analysis, P.S. and H.E.; investigation, I.D.W. and H.E.; data curation, P.S. and H.E.; visualization, P.S. and H.E.; original draft preparation, H.E. and P.S.; writing—review and editing, I.D.W. and H.E.; supervision, H.E. and I.D.W.; final revising, H.E. and I.D.W. All authors have read and agreed to the published version of the manuscript.

**Funding:** This research received no external funding.

**Institutional Review Board Statement:** Not applicable.

**Informed Consent Statement:** Not applicable.

**Data Availability Statement:** The data that support the findings of this study are available from the corresponding author (HE) upon reasonable request.

**Conflicts of Interest:** The authors declare no conflict of interest. The views expressed are strictly those of the authors and do not necessarily represent the positions or policy of their respective institutions.

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
