# Peer review of "Assessing Changes in Land Use/Land Cover and Ecological Risk to Conserve Protected Areas in Urban–Rural Contexts"

_land, doi:10.3390/land12010231_

Round 1

Reviewer 1 Report

The manuscript is good, the authors evaluate the ‘’ Assessing changes in land use/land cover and ecological risk to conserve protected areas in urban-rural contexts’’. It is an interesting and great contribution to the scientific community; however, the material method, discussion and references of the paper should be improved. Still there are many issues present in the manuscript which should be explained properly. The manuscript needs some major revisions as given below:

·         In Abstract, please add methodology and main results.

·         Write LULC changes in Ha and % in Abstract.

·         Set all citied reference in main manuscript according to MDPI style.

·         Write some sentence about remote sensing as well as Landsat related to LULC and ecological risk in Introduction.

·         Line 139, you says that ‘’ 600 samples were examined from the sampled collection’’. Are you collected these point by conducting survey in study area.

·         If you conducted survey in local area, add these points in study area map and write it in methodology section.

·         Table 8, you says that frequency is 35, but you collected 600 sample ?? Please explain the table 8, who can you conduct these analyses and which data used in this.

·         Line 145, you used NDVI team, but you do not find this. If you do not find NDVI, then delete this.

·         The text of this paper in general needs a thorough review, as there are multiple spelling and grammatical errors. Many sentences do not mean any sense. Moreover, there are several sloppy errors that should be fixed.

·         Write the area in Km2 as well as % in abstract.

·         More research background and motivation should be added to the Introduction section. Although, I propose some new papers must be added in the reference list and text which will also help you to make it more intriguing such

https://doi.org/10.3390/atmos13101609

doi:10.24057/2071-9388-2020-117.

·         Your reference style does not match the journal style, please wet this.

·         Resolution of all figures should be improved.

·         In discussion section; Discussion: As per the instruction given by the journal “The findings and their implications should be discussed in the broadest context possible and the limitations of the work highlighted”.

·         Write main results and future recommendation in conclusion.

Overall, the study conducted is interesting but a major revision of the entire manuscript is essentially required for publication in this journal. Hence, I recommend reconsideration after a major revision of the manuscript.  

Author Response

Dear reviewer 

Thank you for the opportunity to resubmit a revised version of the manuscript, and to the reviewers for the valuable feedback to strengthen our manuscript further. Please find in the attach file our detailed response to the comments and remedial actions undertaken.

Best regard,

Reviewer 2 Report

Dear Authors,

The title of the study  Assessing changes in land use/land cover and ecological risk to 2 conserve protected areas in urban-rural contexts” corresponds to its content.

1.      Keywords:  LULC changes; ecological risk assessment; Delphi method; Protected Areas are correct, but I suggest add to word “Iran” and exchange LULC changes” on “Land use/land cover (LULC) changes”.

2.      The total value of work is a valuable contribution. References take 64 publications are cited in the entire article. Literature research well started, but not enough publications from Europe. It is proposed to add the following articles that contain new research in this area, for example: BuÅ›ko, M.; Zyga, J.; Hudecová, Ľ.; Kyseľ, P.; Balawejder, M.; Apollo, M. Active Collection of Data in the Real Estate Cadastre in Systems with a Different Pedigree and a Different Way of Building Development: Learning from Poland and Slovakia. Sustainability 2022, 14, 15046. https://doi.org/10.3390/su142215046

3.      Similarly, the discussion or conclusion should refer to research conducted in this field in other countries from Europe and cited in this publication. Please complete this and the article will be a valuable scientific contribution.

4.      I propose to extend the research to 2021. We already have the year 2022, and in this article, the research has been completed in 2019. Please correct it. The article will be more valuable.

5.      References do not comply with the requirements of the MDPI publication. You should [1, 2, 3] instead (Sinha et al., 2017; Lin et al., 2019; Smeraldo et al., 2020). Please correct it.

6.      References at the end of the article are not numbered and should be. Please correct it.

It should be noted that the whole of the study is cognitive and contains important scientific elements. The article was written at a good academic level. In relation to the above, I express the opinion that the work submitted for review should be published in its entirety after taking into account the comments of the reviewer but not require a review again.

Author Response

(The authors gave the same response as above.)

Reviewer 3 Report

General overview:

The introduction presents a comprehensive summary of the current state of knowledge. Proper planning and execution led to the success of the experiments. This study describes its methodology in detail. High-quality analyses are based on the methods applied. An organized presentation of the study's results with precise graphic and tabular illustrations was provided by the authors. In light of the current literature, I do not value their ability to accurately summarize and interpret the findings. Please see the following for more details.

Specific comments:

I would appreciate your feedback on the following points:

1. It is unnecessary to include this information in the Abstract section. Results from this study should only be presented if they are significant. This section needs to be revised. What can be done to make land use models more accurate and interpretable?

2. It is recommended to change these keywords and avoid conjunctions. Try not to repeat the title of the phrase.

3. It is important to mention the presence of specific main mine waste, including chemical fabric, which may indicate the presence of some positive correlation properties, and clustering is a feature of spatial distribution in these study areas. A professional review of results and presentations is necessary.

4. What factors were considered when determining the location of the study area? For what reason? Is there a particular reason for us the highest total cross-section among the changes in the landscape indices studied? Is there any similar location that could be compared?

5. Is there anything else we can do to improve this data? What are some ways in which this approach could be used in the future? It would be helpful if you discussed this point in the Conclusions and Discussions section.

6. What are some of the benefits and limitations of your approach?

7. References and literature should be added to the Discussion section. This is why the results section needs to be revised. The results should be explained scientifically. Please compare the presented results with other studies. It will increase the scientific value of this publication (if it is possible add more examples).

8. Improve these sentences in the Conclusion section, especially for the destruction of ecosystems leading to high ER.

Constructive feedback:

If you could provide at least one research hypothesis and one directional hypothesis, that would be helpful. It is also imperative to include the results of the management plans and zoning approaches, as well as the qualitative parameters of the output data. As a result of unplanned development using a random sample, the entire chapter "Results" needs to be prepared from the very beginning. There are many input data that need to be addressed in the field of monitoring human activities.

Summary:

There is little discussion of environmental science and ecological issues with the comprehensive idea to evaluate in the article. Most of the discussion focuses on methods that are not yet widely used. This manuscript requires several revisions, as you can see. In the field of planned activities, the work described here is relevant and at the heart of current research approaches.

Author Response

(The authors gave the same response as above.)

Round 2

Reviewer 1 Report

All comments are completed by authors. Now it is ready for publish.

Reviewer 3 Report

You have enhanced the paper and the results; thank you for your feedback. You responded feebly but promptly.